# Advances in Meta-Analysis of the Effects of Grazing on Grassland Ecosystems in China

Xuemin Gong, Yijia Wang, Tianyu Zhan, Chenxu Wang, Changjia Li 🄳 and Yanxu Liu *🄳

State Key Laboratory of Earth Surface Processes and Resource Ecology, Faculty of Geographical Science, Beijing Normal University, Beijing 100875, China; 202121051025@mail.bnu.edu.cn (X.G.)

* Correspondence: yanxuliu@bnu.edu.cn; Tel.: +86-10-58805769

**Abstract:** Grassland ecosystems are among the largest terrestrial ecosystems in China, and grazing, as an important grassland management method, has direct and indirect impacts on grassland ecosystems. Meta-analyses can be used to systematically evaluate and summarize multiple findings from existing studies, but there have been few comparisons of meta-analysis methods. In this review, we summarize the effects of grazing on grassland plants and soil in the existing meta-analysis studies in China from 38 meta-analysis papers. The results show that they have consistent conclusions, such as grazing reduces the aboveground biomass by approximately half, increases the soil pH, decreases the C:N:P ratio, and reduces the number of topsoil microorganisms, but the conclusions of light and moderate grazing index changes vary greatly from study to study. The belowground biomass was generally found to increase, but it slightly decreased in some cases, and the total biomass generally decreased, but it slightly increased in other cases. Vegetation coverage increased during moderate grazing; the soil moisture content was highest for light grazing, and microbial diversity increased at light to moderate levels of grazing. There are also very inconsistent conclusions due to the different datasets and quantities of samples used in meta-analysis studies, as well as variations in the types and scales of grassland areas. The ranges of changes in other indicators were large, especially for the root-shoot ratio and soil carbon. However, changes in the aboveground biomass were generally stable. We suggest subsequent meta-analyses of grazing should further clarify the classification of grassland types and compare conclusions at different scales. Additionally, standardized network analyses are recommended for field manipulation experiments to further improve the accuracy of meta-analysis and reduce the temporal and spatial limitations of existing data.

**Keywords:** grazing intensity; meta-analysis; grassland ecosystem; plant biomass; soil nutrients; soil microorganisms



## 1. Introduction

Grassland ecosystems are among the most important ecosystems in China [1]. Data from the third National Land Survey show that the total area of grassland in China is 2.645 million km$^2$, accounting for 27.5% of China's land area. Grazing is the most important human disturbance type affecting grassland plants and soil systems [2]. The influence of grazing on grassland ecosystems is mainly manifested in the grassland vegetation community [3–6], the soil's physical and chemical properties [7–9], and the changes in microbial indicators in the soil [10–12]. Grazing changes the structure and function of the plant community [13], affects the growth of the plant root system, as coupled with soil trampling and livestock excrement reflux, and promotes the decomposition of plant litter and the nutrient release of plants [14], thus changing the physical and chemical properties of the soil. These changes in the soil environment and underground plant parts greatly affect the underground microbial community [15]. There are six common theories in grassland ecosystems: namely, the grazing optimization hypothesis, the moderate interference hypothesis, the grazing promotion/mitigation ecosystem nutrient cycle hypothesis, the grass

and livestock balance theory, and the grassland agricultural system theory [16]. Many studies have examined six major theories, such as the grazing optimization hypothesis, which states that light grazing and moderate grazing can enable plants to obtain compensatory growth [17]. However, the results on the effects of grazing on plant biomass are different. Yan and Jiang found that under light and moderate grazing conditions, the plant biomass is increased [18,19], but Hao et al. [20] found that the plant biomass of aboveground or underground parts is reduced. Wang et al. also found that light grazing plant biomass decreased by nearly one-fifth, and moderate grazing decreased by nearly one-third [5]. Opponents of the compensatory growth theory argue that grazing has little significance for grassland species [21], suggesting that its theory may lead to the local extinction of plants [22]. Additionally, different conclusions were drawn in different study areas. For example, Zhan et al. [23] analyzed the impact of grazing pressure on the grassland ecosystem of the Inner Mongolia Plateau and found that light grazing has a positive impact on soil fertility and grassland productivity. Jia et al. [24] found that moderate grazing on the Loess Plateau is beneficial for maintaining the growth of the underground parts of plants and that light grazing is beneficial for maintaining soil water and nutrients, while other evidence suggested that grazing improved the alpine grassland ecosystem on the Tibetan Plateau [25]. It is believed that grazing will significantly increase the root-to-stem ratio and reduce the organic carbon of the soil [18]. Thus, there are some controversies in the research results, even in distinguished grazing intensities.

Meta-analysis can provide a systematic evaluation and summary of the results of several studies, so many studies have investigated the effects of grazing on grassland ecosystems using meta-analysis. Some studies have examined the effects of grazing on ecosystem processes and functions in China, such as grassland ecosystem evapotranspiration [26], carbon sequestration [27], plant diversity [27], and plant biomass [28]. Some have conducted meta-analyses of the grazing length and grazing season. Guo et al. found that vegetation coverage showed different responses to short-term and long-term grazing and decreased after short-term grazing but increased after long-term grazing [3]. Liu et al. [7] found that long-term grazing decreased soil respiration more than medium-term grazing. Evidence also found that grazing in the warm season increased the soil temperature more than grazing in the cold season [29]. Due to the great limitations in the scope and data of grazing experiments and because different regions are also affected by different environmental conditions, a further comprehensive analysis is needed of the effects of large-scale grazing on vegetation, soil and microbial systems based on more extensive experimental data. Although it is not uncommon to comprehensively understand the impact of grazing on grassland ecosystems based on meta-analysis, such studies still have limited indicators, and there is no contrast between the conclusions. Therefore, this review tries to sort out the existing progress of meta-analysis and compare the conclusions in these meta-analysis articles while compiling the index changes in the articles according to different grazing intensities. The changing values of the indicators from 38 meta-analysis articles are collected to help us understand the impact of grazing on grassland ecosystems in China, and propose future research directions, hoping to deepen the understanding of the Chinese human–grass–livestock balanced relationship to provide scientific support for the formulation of grassland management policies in China.

## 2. Methodology

Grasslands in China range from high-altitude grasslands on the Tibetan Plateau to low-altitude grasslands in Inner Mongolia. China's grassland ecosystem is mainly divided into four types [30]: typical grassland, meadow grassland, desert grassland, and alpine grassland. The grasslands in China are used for grazing sheep, goats, cattle, yaks, and other livestock. Grassland is the main source of livestock products such as mutton, milk, sheep wool, and cashmere [4].

An advanced literature search using the "Meta * (grazing + grassland)" and "(((TS) (Meta) (AND (TS = (meadow)) OR TS = (steppe)) OR TS = (grassland))) AND TS = (grazing)"

in CNKI and Web of Science, respectively, was performed; the research scope was in China or involved China, and the research content involved meta-analyses of the effects of grazing on grasslands. Overall, 22 and 908 articles were retrieved, respectively. After screening, we filtered out 38 articles and sorted the range of change in different indicators.

The study of the impact of grazing in China began in 2013. Yan et al. [4] conducted a meta-analysis and integrated the impact of different grazing intensities on grassland productivity in China. Meta-analyses have been published since 2016 and reached a maximum of 12 articles published in 2020. There were large differences in the number of articles selected for the meta-analysis, ranging from 10 to 163. The control variables selected by meta-analysis were zero grazing, grazing intensity, grazing and zero grazing contrast, grazing/fence duration, grazing season and grazing form (such as cattle or sheep single grazing or mixed grazing). Most livestock types were cattle and sheep, and sheep units were used in the division of grazing intensity. The research contents include the effects of grazing exclusion/grazing on plant indicators, soil indexes, microbial indicators, the stoichiometric ratio of plant roots and leaves, soil respiration, grassland evapotranspiration, etc.

We collected the values of the indicator changes from 38 meta-analysis articles to obtain the maximum and minimum values for each indicator. Lollipop charts were created using Origin 2021 to represent the range of change in related indicators for grazing effects on plants and soil. Additionally, we created literature review maps and indicator change diagrams forming Figure 1 in PowerPoint. We directly applied the grazing intensity criteria from the original articles in our review. For example, if two studies classified the same grazing intensity differently in terms of sheep units, but both studies classified it as "light grazing," we recorded it as such in our review. This is due to the difference in classification criteria for different study areas. For instance, the sheep unit for light grazing in a desert grassland might be 0.91, while in an alpine grassland, it might be 3.91.

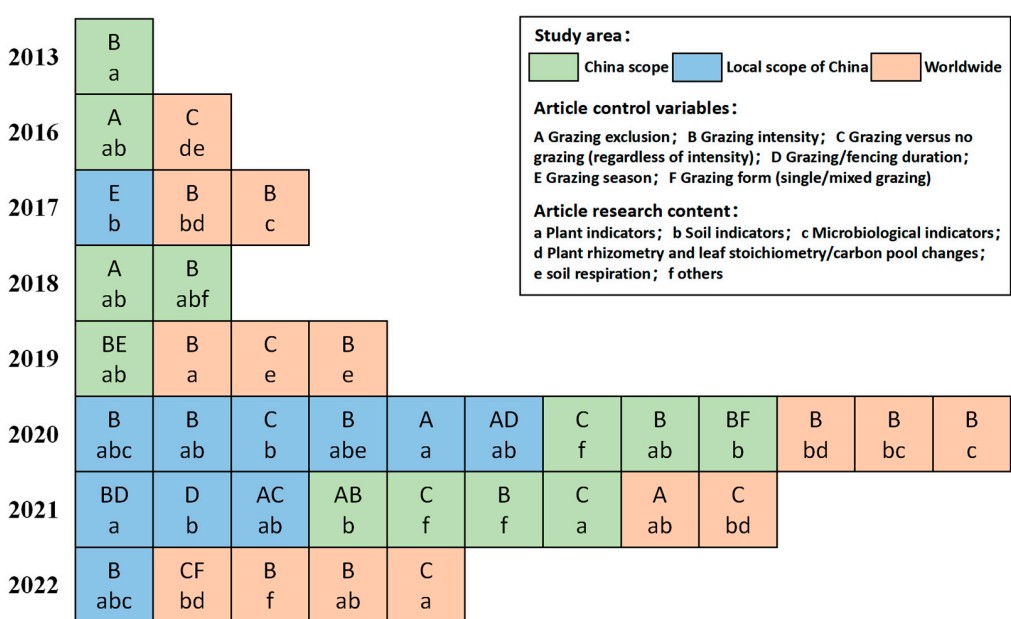

**Figure 1.** A meta-analysis of the effects of grazing/grazing prohibition on grassland ecosystems is available (the number of patches is the number of posts published in the year).

## 3. Results and Discussion

Grazing can significantly alter the ecosystem function and community structure of grasslands, and measures such as overgrazing in China have caused serious grassland degradation, such as in China's largest grassland in Inner Mongolia, nearly half of which has been degraded in the past three decades [31]. The conclusion that grazing affects

grassland is significantly influenced by factors such as grassland type, observation year, time and spatial scale [32,33], grazing conditions, vegetation type, and climatic conditions, among other factors [34]. Furthermore, vegetation production and nutrient cycles depend on soil processes and grazing by livestock on the soil. The effects of plants and excretions change the physical and chemical properties of the soil [35]. The suppression of fertilizer N supply combined with a strong reduction in grazing pressure may not be able to increase in the short term the GHG sink per unit land area of managed grasslands [36], especially in dryland ecosystems [37]. Soil is also an environment for microorganisms to live in. A study on the relationship between the microbial diversity of plants and soil and the versatility of soil nutrients in China found that the importance of soil microbial diversity is greater than that of plant richness under drought conditions [38]. In the following review, we summarize the effects of grazing on plants and soil separately. The effects of grazing on grassland vegetation were analyzed based on biomass and community structure, while the effects on grassland soil were considered from perspectives such as soil structure, soil nutrients, and soil microorganisms. This review summarizes the overall response of vegetation to grazing and the response to grazing at different intensities.

### 3.1. Effects of Grazing on Grassland Biomass

Vegetation biomass is an important support for grassland ecosystem services, such as carbon sequestration, livestock production, and soil and water conservation. At the same time, biomass is directly grazed, supports grassland resources and is a reliable indicator used to measure grassland ecosystem function, productivity, and health [39]. The direct impact of grazing on vegetation is reflected in biomass, and most grazing reduces aboveground biomass (AGB) and total biomass (TB) and increases underground biomass (BGB). The top-down effect of grazing can greatly affect aboveground biomass, thus altering the distribution of the aboveground biomass of plants [40]. At the same time, the changes in AGB and BGB due to grassland grazing depend not only on the external environment and grazing management but also on the species composition [41]. Some evidence shows that grazing increases AGB and BGB [20], but more evidence shows that grazing reduces TB, AGB, and BGB [4,20]. The calculations also lead to very different conclusions: evidence shows that grazing can reduce TB by 58.3%, AGB by 42.8%, and BGB by 23.1%, and significantly increase the root-shoot ratio by 30.6% [4]. Another study suggested that grazing reduced TB by only 14% and AGB and BGB by only 38% and 6%, respectively, but increased the root-shoot ratio by 75% [18]. Although the total biomass decreases with the degradation of grasslands, plants allocate a higher proportion of biomass underground to obtain more nutrients [42]. It has been shown through structural equations that the direct effect of grazing on AGB is −0.45, and the indirect effect is −0.37; additionally, the direct impact on BGB is −0.71, and the indirect effect is 0.19 [43]. Some studies have also found that grazing in early spring has a negative impact on leaf area and photosynthesis for dark respiration in grasslands [44], the combined influence of biological factors and abiotic factors affects the stability of vegetation biomass [40], and grazing intensity and grazing mode are the key factors affecting these differences.

Differences in the response of the grassland biomass to grazing are not limited to grazing intensity but also include growing season grazing and environmental conditions, and there are differences in magnitude in different types of grasslands. Heavy grazing is generally considered to have the greatest negative impact on biomass, followed by light grazing, and moderate grazing has the least negative impact on biomass [19,28] and can even promote vegetation production in arid grasslands [45]. A somewhat opposite opinion argues that although moderate grazing has little effect on BGB, AGB can be reduced by one-fifth, and light grazing is the best choice and has no effect on aboveground and underground biomass [23]. The effect of grazing on biomass is also reflected in the growing season, with nongrowing season grazing significantly increasing AGB, while year-round grazing and growing season grazing significantly reduce AGB [20]. The responses of AGB and BGB to grazing in grassland types are also different. In one study, BGB and TB were negatively

affected by grazing in the following order: desert grassland > meadow grassland > alpine grassland > temperate grassland. Moreover, the negative impact of AGB ranked as follows: meadow grassland > alpine grassland > desert grassland > temperate meadows [28]. Another study showed that the total biomass of desert grassland decreased by 44% after grazing in the middle of the growing season, while that of meadow grassland increased by 40% [3]. Changes in biomass also depend on the combined effects of grazing intensity and environmental conditions: AGB has a secondary relationship with precipitation at light grazing intensities and a linear relationship with precipitation at moderate and severe grazing intensities but does not change with temperature, while the effect of grazing on BGB varies neither with temperature nor with precipitation [4]. The impact of grazing on biomass at different scales is also different; for example, the impact of grazing on grassland biomass in Inner Mongolia is higher than that at the global scale but lower than that at the national scale. In comparing national and global conclusions, grazing has a considerable negative impact on the grassland biomass in China [3]. Grazing intensity affects biomass, which in turn makes the response of the root-to-shoot ratio more significant with increasing grazing intensity, and high grazing intensity leads to a large proportion of subsurface material distribution. As shown in Figure 2, the change in BGB under light grazing is greater than that for AGB, and different grazing intensities will reduce AGB, while the change in the root-to-shoot ratio is greatest under heavy grazing. This large difference in grazing intensity may be caused by different vegetation types in different grasslands.

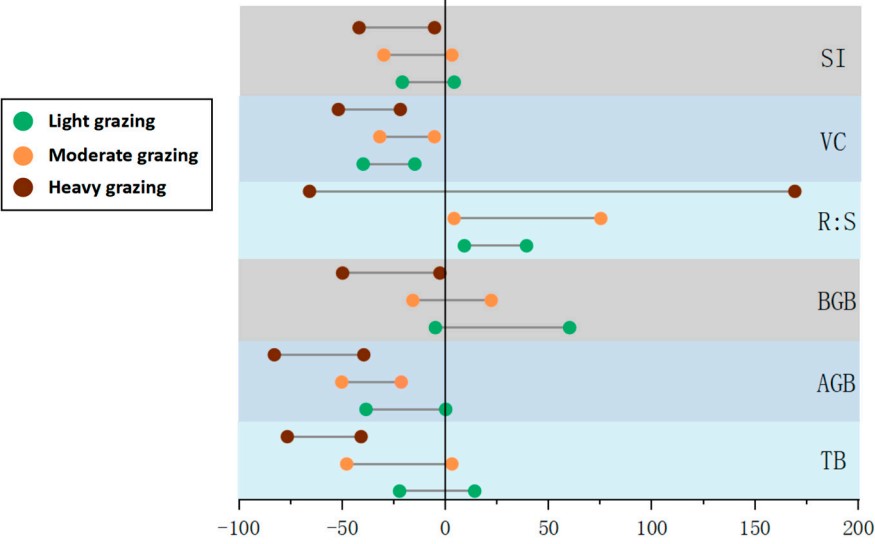

**Figure 2.** Variation ranges of plant indicators under different grazing intensities. TB: total biomass; AGB: aboveground biomass; BGB: underground biomass; R:S: root-to-shoot ratio; VC: vegetation coverage; SI: Shannon—Wiener index.

### *3.2. Effect of Grazing on Grassland Community Structure*

The stability and organization level of the vegetation community directly affect the stability of grassland ecosystems, and grazing may lead to different changes in biodiversity, ecological function and the relationship between the two [44]. The effects of grazing on community structure are mainly associated with vegetation coverage, community height, and species richness [34], while grazing leads to increased plant species richness and reduced vegetation coverage through negative effects on biomass [5]. It is generally believed that grazing reduces the coverage, height and density of most vegetation [5,46], and the effects are variable depending on the different regions and scales. On the Loess Plateau, it is believed that the species diversity and richness of desert grassland decreased by 48.8% and 27.8%, respectively, under the influence of grazing [24], while in the alpine grassland in the Qinghai–Tibet Plateau, the species diversity increased by 7.3%, and the species richness increased by 9.9% [25]. The influence of grazing on community structure

is influenced by environmental factors and grassland types. Some scholars believe that the influence of grazing disturbance in grassland ecosystems is overestimated. Ren et al. [47] found that during grazing periods, precipitation and temperature can have larger effects than the actual grazing disturbance, using simple linear regression to test the correlation between the biomass of the target species and diversity parameters in Inner Mongolia. The duration of grazing also affects grassland ecosystem changes. In one study, short-term grazing reduced the vegetation coverage of desert grassland by 50%, and medium-term grazing reduced the coverage of desert grassland and meadow grassland by 65% and 15%, respectively, while the typical grassland response was not obvious [3]. Grazing has different effects on the composition of different vegetation communities, with no effect on annual herbs but a reduction in the richness of perennial herbs. Shi et al. [48] conducted a global meta-analysis based on 483 paired comparisons from 62 experimental studies to evaluate grazing effects on soil seed banks. They divided grazing intensity into four categories and found for soil seed bank abundance, grazing had no effect on annual herbs but decreased the abundance of perennial herbs, and compared with functional groups, grazing has positive effects on weed richness and negative effects on grasses.

The response of grassland to grazing under the most moderate interference hypothesis involves the vegetation diversity, species richness index, and vegetation density [5,49,50], which significantly increase under light grazing and moderate grazing; additionally, when heavy grazing significantly decreases, light grazing and moderate grazing had similar effects on diversity [51], and the influence of the vegetation density gradient was significant [52]. Some studies also found that these indicators do not fit the moderate interference hypothesis and decrease with an increase in grazing intensity [53,54]. However, this is only valid for typical grasslands. The diversity of desert grassland makes them more sensitive to grazing intensity, with diversity decreases of 8% for light grazing, 11% for moderate grazing, and 19% for heavy grazing; conversely, the diversity of meadow grassland is not affected by light or moderate grazing and only decreases by 10% [5] during heavy grazing. A study on the effect of grazing intensity on the water utilization rate of grassland vegetation communities in northern China showed that heavy grazing was significantly reduced, while the effect of light grazing was not significant; moreover, moderate grazing improved the water utilization rate of vegetation [55]. Under the influence of grazing, the AGB of vegetation also varied in terms of the relationship with vegetation community characteristics in different grassland types. Generally, grazing can lead to a linear relationship between AGB and vegetation coverage, height, and species abundance, but changes in community composition and environmental conditions can turn these linear relationships into unimodal relationships, such as for AGB and species richness in meadow grassland, AGB and height in typical grasslands, and AGB and coverage in desert grassland [34]. Temporal asynchrony is also an important factor in regulating the stability of grazing in different grasslands. Cleveland et al. [56] found that species asynchrony plays a key role in typical and meadow grasslands, while desert grassland is affected by species asynchrony, richness and species stability. Moreover, light grazing and moderate grazing may be positive or negative for different types of grassland, but heavy grazing will certainly reduce the diversity of all types of grasslands, simplify the vegetation community structure, and destroy the stability of grassland ecosystems.

### 3.3. Effect of Grazing on the Grassland Soil Structure and Environment

Most grassland soil is compacted by livestock [57,58], which will increase the soil bulk density, soil pH, and soil temperature and reduce soil moisture [59]. Grazing can also significantly change the mechanical composition of soil, and decrease the mucus content in soil [7]. The effect of grazing on the physical properties of the soil is influenced by precipitation and temperature. When the temperature increases, the positive effect of grazing on the soil bulk density and soil temperature and the negative effect on soil moisture will become increasingly obvious. When precipitation is greater than 500 mm, the response of soil temperature to grazing will become less obvious [60]. Soil and vegetation

are the two major parts of grassland ecosystems, and the physical properties of the soil are also clearly associated with plant biomass. BGB is positively correlated with the soil water content and pH and negatively correlated with conductivity and soil bulk density; AGB is positively correlated with the soil water content and conductivity and negatively correlated with the soil bulk density and compaction [7]. However, it remains to be determined whether the correlation between them is direct or indirect. The changes in soil physical traits also affect the distribution of plant communities, and soil firmness and porosity are affected the most [61]. Moreover, grazing also has a temporal accumulation effect on soil compaction, and the longer the grazing time, the greater the soil compaction [62]. In a study on the Loess Plateau, it was found that grazing reduced the soil water content of desert grassland by 17.2% and increased the pH by 1.3% [24]. On the Qinghai–Tibet Plateau, grazing reduced the soil water content of alpine grassland by 20.8% [25], and in a national-scale study, grazing reduced the soil water content by 8.2% [20]. Although the values were obtained at different scales, the reduction trend was the same.

The main impact of grazing is on the physical properties of the topsoil. With increasing grazing intensity, the soil bulk density, porosity, pH and water content decreased [63,64]. Some evidence shows that the soil water content and porosity are also consistent with the moderate interference hypothesis; moderate grazing can increase the soil water content from 10 to 20 cm by 22.5% and reduce the soil bulk weight by 9.7% [65], while the soil water content from 0 to 10 cm varies linearly with grazing intensity [66]. The particle size composition from 0 to 10 cm is particularly significant, with a sand content of 10% to 30% [67]. Under the effects of light, moderate, and severe grazing, the soil bulk density from 0 to 10 cm increased by 5%, 8%, and 11%, respectively, but the effect on deep soil bulk density was limited [68]. A study on the effects of grazing and precipitation on alpine grassland in the Qinghai–Tibet Plateau found that when the 0–5 cm soil layer was not deficient, light, moderate, and severe grazing by yaks had a greater influence on the soil bulk density than did grazing with Tibetan sheep, and in the 5–10 cm soil layer, the effects of the yaks and Tibetan sheep increased, but the difference was not significant [69]. Both grassland type and grazing species had a significant impact on the structure of the 0–10 cm soil layer, and the deeper the soil layer was, the lower the impact. Unlike alpine grasslands, after short-term grazing in meadow grasslands, soil bulk weight does not change significantly under different grazing intensities, but soil moisture is minimized during moderate grazing [70]. The corresponding soil moisture has different grassland types. For example, a study in alpine grasslands found that soil moisture rises first and then decreases with increasing grazing intensity, which is completely contrary to the conclusion for meadow grasslands [71]. According to 37 meta-analysis studies, the soil pH ranged from 0.5% to 1.0%, with values of 0.6–4.1% and 1.0–5.2% for moderate and heavy grazing, respectively; additionally, the soil water content ranged from 4.8–11.0%, with values of −6.3–2.6% and −23.9–10.8% for moderate and heavy grazing, respectively [9,20,72]. These values correspond to soil depths generally from 0 to 50 cm, with most from 0 to 10 cm.

*3.4. Effect of Grazing on Soil Nutrients*

Grazing reduces grassland biomass and litter through feeding and trampling, changing the soil structure, reducing organic matter sources in the soil, and accelerating the loss of nutrients from the soil itself [73]. However, nutrients from grazing excreta are returned to the soil through decomposition, which promotes the circulation of soil nutrients [74]. Grazing had the greatest impact on the topsoil, with an effect from 0 to 10 cm higher than that from 10 to 40 cm and decreasing with increasing soil depth [75]. Soil nutrient responses to grazing varied across grassland types. Under the influence of grazing in alpine grassland, soil organic carbon decreased by 13.7%, soil total nitrogen decreased by 12.7%, soil total phosphorus decreased by 11.6%, soil effective phosphorus decreased by 7.7%, and the soil carbon to nitrogen ratio decreased by 3.4%. Moreover, the response rate of soil organic carbon was positively correlated with the response rates of total nitrogen, quick phosphorus, the carbon–nitrogen ratio, and the carbon–phosphorus ratio [7]. However,

some evidence shows that grazing can increase the carbon and nitrogen reservoirs of grassland ecosystems [76], while the effects of grazing on alpine grasslands amplify the uncertainty of change [77]. In a typical grassland, grazing had no significant effect on soil total carbon and soil quick nitrogen, though the soil total nitrogen in the 30–40 cm soil layer was significantly affected by grazing [78]. In desert grassland, the change in total phosphorus was most pronounced, probably due to changes in litter decomposition [79]. Soil net nitrogen mineralization rates also responded differently to grazing, with increases in alpine meadows and decreases in meadow and desert grasslands [80]. The rates in 0–30 cm and 0–10 cm soil layers were higher than those in a free grazing area [81]. In addition to the above indicators, organic matter, nitrate nitrogen, ammonium nitrogen, total potassium, quick potassium, and other indicators will decrease with grazing, and there is a significant correlation between the total amount of nutrients and plant biomass [82]. Grazing has different effects on the soil nitrogen and phosphorus cycles. Grazing reduces the soil nitrogen reservoir, but it increases the soil phosphorus reservoir. Other conditions also affect the response of grassland soil nutrients to grazing, and if the litter biomass is reduced and waste production is increased, grazing will increase the carbon loss in the atmosphere [83]. When the temperature is greater than 0 °C, grazing will increase the soil carbon reservoir; if precipitation is less than 0 °C, grazing soil phosphorus will decrease by 4.2%, but the soil phosphorus will increase by 2.3% at 400–800 mm, though with increasing elevation, the soil carbon reservoir reaction to grazing changes from positive to negative, and the soil carbon phosphorus ratio is the opposite [84].

In different soil layers of grassland ecosystems, the intensity of grazing can affect the contents of different nutrients. Grazing intensity mainly affects the soil organic matter and total nitrogen in the surface and parent layers from 0 to 5 cm in the soil, while the contents of total phosphorus, total potassium, and inorganic carbon are mainly affected in the 10–15 cm soil transition layer; additionally, the influence on nitrate and ammonium nitrogen is mainly in the non-surface and parent layers [85]. The soil organic matter in the 0–10 cm soil layer did not change significantly under different grazing intensities, while that in the 10–20 cm soil layer changed more due to moderate grazing than to light grazing, and the total nitrogen in the 10–20 cm soil layer was highest under moderate grazing [86]. Researchers found that mixed grazing of cattle and sheep in alpine grassland could increase soil organic matter, soil total nitrogen and total phosphorus content, with no significant effect on quick nitrogen [87]. In addition, a study of grazing during the growing season in an alpine grassland found that soil organic carbon was negatively correlated with grazing intensity, with no significant effect on quick nitrogen [88]. In general, heavy grazing has a large negative impact on soil stoichiometry and limited light and moderation effects; additionally, light grazing helps promote the recovery of soil nutrients in grassland ecosystems [23,58,89]. Evidence shows that the soil carbon content under light grazing increased largely because of the increase in the soil carbon content at depth, although the surface soil carbon content decreased. So light grazing changes the allocation of carbon inputs and promotes the accumulation of carbon in the soil [19]. From the perspective of soil respiration, some scholars have suggested that grazing can reduce soil respiration by 12.4% and autotrophic respiration by 15.6% [90]. However, some evidence shows that light grazing and moderate grazing have no negative effect on soil respiration, and only heavy grazing reduces the absorption of soil methane by 36.5% [91,92]. As shown in Figure 3, the variation range of soil carbon was the largest under different grazing conditions. Light and moderate grazing may increase organic carbon but still reduce the total carbon pool. The soil nitrogen pool may increase under light grazing conditions, and moderate and heavy grazing will certainly reduce it. In summary, light grazing has a certain role in promoting the recovery of grassland soil.

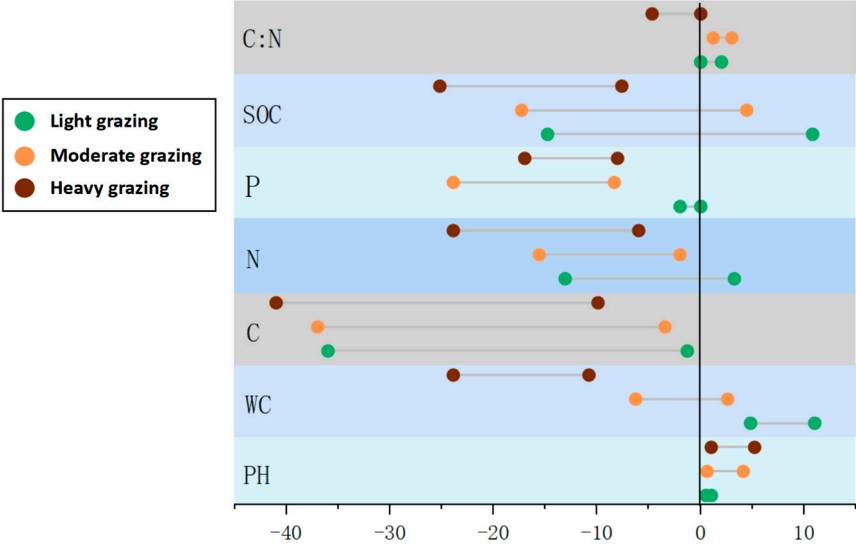

**Figure 3.** Variation ranges of soil indicators under different grazing intensities. WC: soil water content; SOC: soil organic carbon.

### 3.5. Effect of Grazing on Soil Microorganisms

Microorganisms are involved in the chemical cycles of plants and soil and are a significant component of the soil ecosystem that cannot be ignored. Grazing can alter the soil structure and environment, which in turn can affect the living conditions of soil microbes [93]. Grazing can maximize the changes in the number, structure, and distribution of microbial communities in the soil, resulting in the emergence of more endemic microbial communities. Additionally, microorganisms, such as bacteria and fungi, selectively use different forms of soil elements, with common sensitivities to ammonium and nitrate nitrogen [94]. A study conducted on simulation of the response of soil microbial carbon and Nitrogen to grazing found that grazing can reduce the carbon content of soil microorganisms, but it had no significant effect on the nitrogen content; moreover, grazing can reduce the activity of microorganisms at the end of growth [95]. Grazing also reduces the abundance of bacteria and fungi, increasing their ratio and thus increasing the carbon sequestration potential [24]. In a study of grazing-affected microbial communities and soil respiration, grazing reduced soil microbial, bacterial, and fungal communities by 11.7%, 8.9%, and 11.5%, respectively, and the results varied significantly in different soil layers, by 3.1% and 4% from 0 to 10 cm and at depths greater than 30 cm and by 8.6% and 15.9% at 10–20 and 20–30 cm, respectively [11]. Different topographies, timings, and grassland types can also influence microbial responses to grazing. For example, grazing on the flat ground increases the microbial phosphorus content, though the effect on a slope was not significant [96], and compared to grazing in June and August, there was a higher microbial mass carbon content from grazing in July and September for cattle [97]. In addition, the quantities of bacteria and actinomycetes were lower in grazing than in the grassland of Baikal pinegrass (Stipa Baicalensis Roshev) [98], and desert grassland bacterial communities responded to grazing faster than typical grassland and meadow grasslands [99]. Microbial responses to different grazing practices are also different, and a study in Yunnan found that Tibetan pig grazing affected ammonia-oxidizing archaea more negatively than did yak grazing [77].

The sizes of microorganism communities, the carbon, nitrogen, and phosphorus contents and the populations of some enzymes are highly sensitive to the grazing intensity, grazing length and grazing system. Grazing reduced the microbial community size by 11.7% on average, with light and moderate grazing increasing the community size by 5.0% and 1.9%, respectively, and heavy grazing resulted in an 18.1% decrease [11]. A study in an alpine grassland found that the diversity index of soil fungal communities was not

significantly different among different grazing intensities, but the richness was highest for light grazing and roughly the same under moderate and heavy grazing [100]. A study in Shanxi Province found that light grazing reduced the mean abundance of both bacteria and fungi, and the abundance of fungi under the conditions of light and moderate grazing only significantly increased in September [101]. Considering the large difference in microbial abundance and the significant influence of many factors, a study analyzed the abundance change in dominant taxa and showed that most of the dominant taxa did not change because of the different grazing systems, and continuous grazing had a greater negative impact on microorganisms than rotational grazing [102]. In terms of the microbial biomass of carbon and the biomass of nitrogen, light and moderate grazing increased the microbial biomass of carbon and nitrogen [2,103], which were significantly correlated with soil total carbon and total nitrogen; additionally, the microbial biomass of carbon was positively correlated with total nitrogen and soil soluble carbon [104]. The response of microbial biomass carbon varied with grazing gradients and time, with the effects of light grazing and heavy grazing decreasing over time and the effects of moderate grazing decreasing first and then increasing; no such interaction was observed for the biomass nitrogen [62]. Ammonia-oxidizing bacteria displayed the lowest abundance during moderate grazing, with an 18.1% reduction, and the diversity indices of both ammonia-oxidizing bacteria and ammonia-oxidizing archaea decreased with increasing grazing intensity [105]. The total amount of phospholipid fatty acids in soil microbial communities was positively correlated with grazing intensity, which was 12.4% and 37.3% higher than that for moderate and heavy grazing, respectively [106]. We believe that these results and expanding the scope of meta-analysis can help us consider a more comprehensive approach to ecological conservation and grassland livestock management.

## 4. Conclusions

We selected the indicators that had two or more data appear in 38 articles for sorting out in Figure 4. They appeared 6–29 times in the process of article statistics. As shown in Figure 4, although the effect of grazing on some indicators of the grassland ecosystem varied, the response of aboveground biomass was negative during light grazing and moderate grazing, and the response of underground biomass was positive and negative depending on the conditions. Compared with that during light grazing, the change in vegetation coverage during moderate grazing increased, and the increase in the root-shoot ratio conformed to the changes in aboveground and belowground biomass. The change in plant diversity also decreased during light grazing and moderate grazing. The impact on the soil environment can be gradually seen in the soil under the interference of grazing. Grazing changes the pH, and an increase in grazing intensity has a drastic effect. For instance, the soil water content under light grazing exhibited the greatest change, and the range under moderate grazing varied from positive to negative. Further, carbon, nitrogen and phosphorus changes showed negative responses, and the biggest influence was soil carbon, where the organic carbon content under light and moderate grazing conditions may increase or decrease. The drastic difference in the carbon and nitrogen ratio is only seen during heavy grazing, while this ratio increased in cases with light and moderate grazing, though the change was not significant. The ratio varied between light and moderate grazing but showed a negative response and a wide range of changes during heavy grazing. Changes in grazing affecting biomass may fit the grazing optimization hypothesis in terms of total biomass, but the changes are generally reduced in terms of aboveground biomass [18,21].

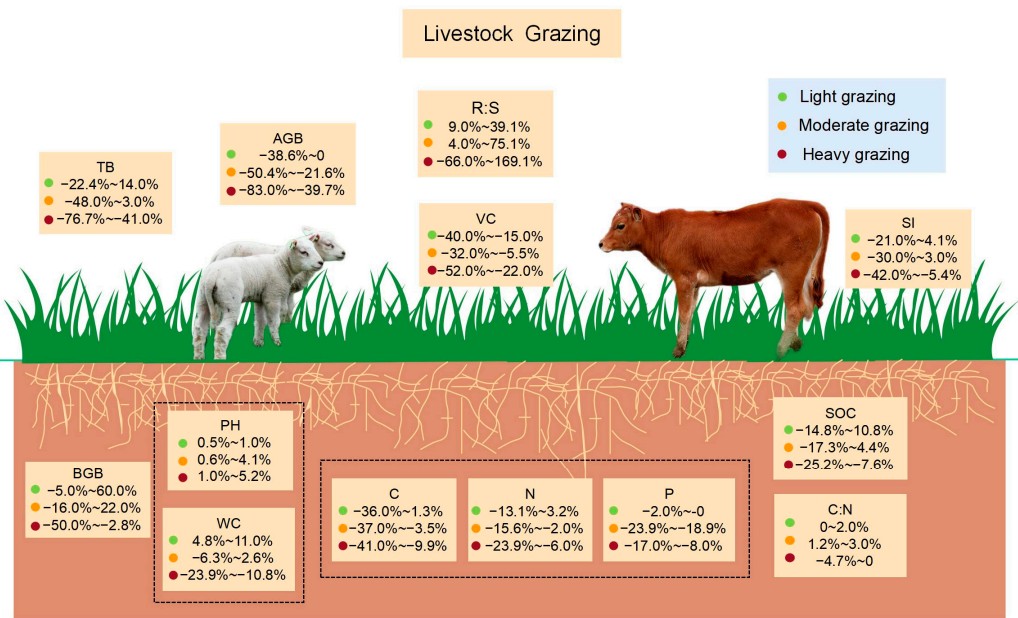

**Figure 4.** Ranges of response values of grassland ecosystem indicators to grazing. TB: total biomass; AGB: aboveground biomass; BGB: underground biomass; R:S: root-to-shoot ratio; VC: vegetation coverage; SI: Shannon—Wiener index; WC: soil water content; SOC: soil organic carbon.

The large ranges of indicator values and controversial conclusions may be due to the use of different study sites and study scales, as well as different divisions of grazing intensity and soil indicators, which are influenced by environmental factors such as temperature, precipitation, terrain and elevation. These issues are mainly related to the large differences in the quality and quantity of previous articles selected for meta-analysis. In addition, whether grazing will promote or slow the soil nutrient cycle requires further statistical analysis. In the existing analyses, only the response ratio of soil nutrients was included, and the effect on the nutrient cycle was not considered. In addition, this review only considers the four simple types of grasslands and does not classify more grassland types according to multiple classification criteria. China has a vast territory, and the conclusions in more detailed specific areas of grassland need to be further explored.

## 5. Future Research Prospects

From the effects of grazing on grassland already involved in the content of the meta-analysis papers, we believe that further meta-analysis to study the impact of grazing on grassland ecosystems should mainly focus on five factors: refining control variables, increasing analysis indicators, expanding spatial and temporal scales, quantifying environmental impacts, and exploring and establishing system networks. At present, many studies have focused on the influence of grazing intensity [23,24,26,55,80], and there are few studies on the grazing season [7], grazing duration [20], livestock type [25], and grazing mode [99]. The effects of grazing on grassland vegetation were mostly focused on the grassland biomass [4,28,44], diversity [5,27,91], richness [24,25], the root-shoot ratio [18], vegetation coverage [3,107], the carbon content of the roots [59,84], and other indicators. Studies of the soil response to grazing have mostly focused on soil physicochemical property indexes, including those related to the soil environment [19], bulk density [27], density [20], soil respiration [59,89] and changes in the carbon, nitrogen and phosphorus indexes of soil nutrients [7,18,25,99,107]. Research on microbial responses to grazing has mostly concentrated on the microbial community area [11], number [27], abundance [12,24], and changes in carbon, nitrogen and phosphorus indicators of microbial biomass [58,83]. However, there is currently no nationwide meta-analysis available on the effects of grazing on plants and nutrient stoichiometry in China. [84]. Compared with those on plants and soils, there

are still few articles on the microbial responses to grazing, and it is necessary to strengthen knowledge of the impact of grazing on the carbon pool of grassland ecosystems in China in the context of a carbon-neutral strategy [108]. In meta-analyses, some interactions can be explored to identify the compound effects of grazing and other disturbances [109], such as changes in temperature and precipitation, which can change the proportion of grazing effects in a grassland ecosystem [7]. Considering the plant—soil-vegetation reciprocal feedback relationship in grassland ecosystems and the coupled plant—soil-microbial system changes, we can try to establish a relationship network among indicators and the environment for composite impact analysis. Otherwise, how to restore the grassland degraded due to overgrazing is an important problem to the world, some scholars proposed measures of banning grazing to restore the grassland naturally [110,111], and some suggested that moderate grazing can speed up the recovery of the grassland [2]. For the grassland in different areas, how to manage the grazing intensity, pattern and duration remains to be further studied and organized.

Finally, meta-analysis, as an integrative analysis tool, plays a positive role in the synthesis of long-sequence and multiscale studies. Many studies on how grazing affects grassland indicators have been performed, but there is a lack of quality assessment of these articles for a more accurate analysis of previous conclusions [112]. Due to the limitations of experimental scope in field trials, coupled with differences in experimental design methods and determination frequency, there can be increased uncertainty in overall results [113]. Additionally, sufficient data is required to assess visual, spatial, and temporal changes and make subsequent predictions [114]. Therefore, standardized networked experiments are needed to further improve the quality of research [115], which means in subsequent grazing control experiments, the whole of China or the world could use the same experimental standard to work.

**Author Contributions:** Conceptualization, X.G., Y.W. and T.Z.; validation, Y.L., C.L. and C.W.; investigation, X.G.; data curation, X.G.; writing—original draft preparation, X.G.; writing—review and editing, Y.L.; visualization, X.G.; supervision, Y.W. All authors have read and agreed to the published version of the manuscript.

**Funding:** This research was funded by the National Natural Science Foundation of China (42171088, 42007052), the Science and Technology Project of Inner Mongolia Autonomous Region, China (NMKJXM202109), and the Fundamental Research Funds for the Central Universities of China.

**Institutional Review Board Statement:** Not applicable.

**Data Availability Statement:** Not applicable.

**Conflicts of Interest:** The authors declare no conflict of interest.

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
