# Peer review of "Advances in Meta-Analysis of the Effects of Grazing on Grassland Ecosystems in China"

_agriculture, doi:10.3390/agriculture13051084_

Round 1

Reviewer 1 Report

The manuscript describes the effects of grazing on grassland ecosystems in China. This topic is relevant due to the rapid development of animal husbandry in all parts of the world.

Comments for Authors:

Despite this being a review, it is recommended to follow the usual article structure: introduction, methodology, results and discussion, conclusion. It means to re-name the section 2 of the article (from “Statistical analyses of the literature” to “Methodology”) and putting the sections 3 and 4 together into “Results and discussion”. This is only a recommendation; I believe, such changes will improve the article.

The methodology is not clear enough. Please provide a detailed description of the statistical analysis.

There is a lot of information about “light grazing”, “moderate grazing”, “heavy grazing”. But there are no criteria mentioned. How can you conclude if there is light or heavy grazing? Please, add the such information into the "Methodology".

Author Response

Dear reviewers,

Thanks very much for taking your time to review this manuscript. We thank the reviewers for the time and effort that they have put into reviewing the previous version of the manuscript. We really appreciate all your comments and suggestions! Based on the instructions provided in your letter, we uploaded the file of the revised manuscript. Please find our itemized responses below and revisions in the re-submitted files. The comments are reproduced and our responses are given directly afterward in a different font.

We would like also to thank you for allowing us to resubmit a revised copy of the manuscript. We hope that the revised manuscript is accepted for publication in the Agriculture.

Sincerely,

Xuemin Gong.

Reviewer 2 Report

Dear editor

The structure of paper represent a review under a new analysis. The Grazing is a priority topic in conservation management, however the additive value of paper including a new analysis and new approach.  Accordingly Author(s) should be pay attention to some key comments including:

Title

Grazing or over-grazing?

Abstract

Should be revised after the main text revision

Introduction  

What are the additive fitness of this study?

New analysis?

New approach?

Collection and comparison?

and etc.

The structure of manuscript including a mix of review in frame a new analysis, so more powerful introduction seems to be necessary to improve

Hypothesis

Necessity

justification

aims

to justify the importance or study or represent the vacancy of similar studies 

Authors should be provide a suitable structure for material and methods including:

Study area: comprehensive description on ecology and physical geography of studied area

Methods: software, analysis and etc.

Results

Authors should be classify the main titles to study area, results and discussions or conclusion

The cited titles (e.g. effects of grazing on....) should be considered as subtitles of result

Discussion

What are the main achievements the study in frame of

Conservation ecology

Ecosystem management

Environmental plans and etc.?

Finally the manuscript including valuable concepts and new analysis, regardless showing some shortcomings that must be corrected in detail.

Decision: Major Revision

Best Regards

Dear editor

The structure of paper represent a review under a new analysis. The Grazing is a priority topic in conservation management, however the additive value of paper including a new analysis and new approach.  Accordingly Author(s) should be pay attention to some key comments including:

Title

Grazing or over-grazing?

Abstract

Should be revised after the main text revision

Introduction  

What are the additive fitness of this study?

New analysis?

New approach?

Collection and comparison?

and etc.

The structure of manuscript including a mix of review in frame a new analysis, so more powerful introduction seems to be necessary to improve

Hypothesis

Necessity

justification

aims

to justify the importance or study or represent the vacancy of similar studies 

Authors should be provide a suitable structure for material and methods including:

Study area: comprehensive description on ecology and physical geography of studied area

Methods: software, analysis and etc.

Results

Authors should be classify the main titles to study area, results and discussions or conclusion

The cited titles (e.g. effects of grazing on....) should be considered as subtitles of result

Discussion

What are the main achievements the study in frame of

Conservation ecology

Ecosystem management

Environmental plans and etc.?

Finally the manuscript including valuable concepts and new analysis, regardless showing some shortcomings that must be corrected in detail.

Decision: Major Revision

Best Regards

Author Response

(The authors gave the same response as above.)

Reviewer 3 Report

The manuscript discusses on two points:

  1) Advances in meta-analyses of grassland ecosystem in China 

  2) Effect of livestock grazing on grassland ecosystem in the same country

The authors can find specific comments on the manuscript highlighted in yellow color.

General Comments;

1) The topic on effect of livestock grazing on grassland ecosystem is very important and several scenarios raised in the manuscript. As over half of the earth surface covered with grassland/rangeland and considering the economic importance of this ecosystem for the livelihood of the society, particularly in developing nations, grazing management is very important. To this end, the manuscript tried to exploit existing grassland-based published articles.

2) The issues of "Advances in meta-analysis" seems not relevant to the topic. The authors better to stick on the analyses of effect of livetsock grazing on the grassland ecosystem. I didn't see much scientific contribution of this section except the number of papers reviewed on each meta-based works. What is presented on "Statistical Analyses of the Literature" on #2 as an indicator of the advances in meta-analyses is not enough.

3) The effect of continuous grazing leading to overgrazing is perceived as the disturbance factor of grassland ecosystem, particularly soil degradation is key issue here. The authors better to include/show rehabilitation efforts of overgrazed/degraded grasslands as they became difficult and will take a number of decades to bring back.  Linking the effect of overgrazing on grassland ecosystems with efforts being made to rehabilitate those degraded grassland is a better way out. 

In addition, different grassland forage species (grasses/legumes E.g. Timothy vs Clover) vary in their response to different grazing types and livetsock species involved. Despite authors indication in their conclusion as this points are missing, its very crucial part in grassland grazing management. 

Some of the references not cited properly; E.g. According to Ha et al. [  ], ..... in such type of citation you simply put " Ha et al", which is wrong.  

In some areas the sentences lack coherence and disconnected with next sentences; E.g. on Section 3.2. on the effect of precipitation and temperature.

In some areas the language used needs to be revised. I indicated on the manuscript. 

Author Response

Dear reviewer,

Thanks very much for taking your time to review this manuscript. We thank the reviewers for the time and effort that they have put into reviewing the previous version of the manuscript. We really appreciate all your comments and suggestions! Based on the instructions provided in your letter, we uploaded the file of the revised manuscript. Please find our itemized responses below and revisions in the re-submitted files. The comments are reproduced and our responses are given directly afterward in a different font.

We would like also to thank you for allowing us to resubmit a revised copy of the manuscript. We hope that the revised manuscript is accepted for publication in the Agriculture.

Sincerely,

Xuemin Gong.

Round 2

Reviewer 2 Report

Dear Editor

The errors are corrected. The manuscript is acceptable

Best Regards

Author Response

Dear reviewer,

Thanks again to the two reviewers and editors for their affirmation of the article. The comments of you contributed to the further optimization of the manuscript  and increase the scientificity and rigor of the article.Thank you for your efforts to revise the article!

Sincerely,

Xuemin Gong.

Reviewer 3 Report

I have no comments for the authors. 

Author Response

(The authors gave the same response as above.)
